

# Factors associated with non-attendance at appointments in the gastroenterology endoscopy unit: a retrospective cohort study

Hasan Yılmaz[1,2] and Burcu Kocyigit[2]

[1] Department of Gastroenterology, Kocaeli University, İzmit, Kocaeli, Turkey
[2] Department of Internal Medicine, Kocaeli University, İzmit, Kocaeli, Turkey

## ABSTRACT

**Background and Aims:** Gastrointestinal (GI) endoscopy is a limited health resource because of a scarcity of qualified personnel and limited availability of equipment. Non-adherence to endoscopy appointments therefore wastes healthcare resources and may compromise the early detection and treatment of GI diseases. We aimed to identify factors affecting non-attendance at scheduled appointments for GI endoscopy and thus improve GI healthcare outcomes.

**Methods:** This was a single-center retrospective cohort study performed at a tertiary hospital gastroenterology endoscopy unit, 12 months before and 12 months after the start of the COVID-19 pandemic. We used multiple logistic regression analysis to identify variables associated with non-attendance at scheduled appointments.

**Results:** Overall, 5,938 appointments were analyzed, and the non-attendance rate was 18.3% (1,088). The non-attendance rate fell significantly during the pandemic (22.6% $vs.$ 11.6%, $p < 0.001$). Multivariable regression analysis identified the absence of deep sedation (OR: 3.253, 95% CI [2.386–4.435]; $p < 0.001$), a referral from a physician other than a gastroenterologist (OR: 1.891, 95% CI [1.630–2.193]; $p < 0.001$), a longer lead time (OR: 1.006, 95% CI [1.004–1.008]; $p < 0.001$), and female gender (OR: 1.187, 95% CI [1.033–1.363]; $p = 0.015$) as associated with appointment non-attendance.

**Conclusions:** Female patients, those undergoing endoscopic procedures without deep sedation, those referred by physicians other than gastroenterologists, and with longer lead time were less likely to adhere to appointments. Precautions should be directed at patients with one or more of these risk factors, and for those scheduled for screening procedures during the COVID-19 pandemic.

## INTRODUCTION

The endoscopy unit is at the center of gastroenterology practice, and various vital interventional procedures, especially early-stage gastrointestinal cancer screening and treatment, are performed there. Access to gastrointestinal endoscopy is limited in many countries due to the scarcity of suitably qualified personnel and lack of technical

Corresponding author
Hasan Yılmaz,
hasan.yilmaz@kocaeli.edu.tr

equipment. Failure to attend appointments arranged with endoscopy units may compromise healthcare effectiveness. Non-attendance rates are reported to vary widely and this has been linked to a number of factors (*Wong, Zhang & Enns, 2009*; *Chang, Sewell & Day, 2015*; *Nash et al., 2006*). Non-attendance at the appointment was reported to be 3.6% in a study in which patients were referred by a specialist or in an emergency. In contrast, 27% of patients did not attend appointments where non-attendees were of advanced aged, had a prolonged lead time, or were homeless or unemployed. The non-attendance rate was as high as 67% in non-urgent endoscopies and for screening colonoscopies. When appointments are not honored, the overall waiting time increases and patient satisfaction is likely to decrease (*May et al., 2017*). Additionally, non-attendance at scheduled appointments has been reported to increase emergency department admissions and hospitalizations (*Hwang et al., 2015*; *Nuti et al., 2012*). Identifying the determinants of non-attendance and taking appropriate measures against them will lead to a more effective use of healthcare resources.

Several factors, such as older age, male gender, being single, not having health insurance, having a low income, appointment in the winter, and longer travel distance to hospital, have all previously been identified as being associated with lower attendance at appointment (*Partin et al., 2016*; *Shuja et al., 2019*; *Vutien et al., 2019*; *Nayor, Maniar & Chan, 2017*). Targeting identified non-attendance factors using methods, such as patient navigation programs, nurse phone calls, and over-booking appointments, was found to reduce non-attendance rates. These interventions increased the efficiency of the endoscopy unit and treatment success and provided economic benefits (*Nayor, Maniar & Chan, 2017*; *Childers et al., 2016*; *Reid et al., 2016*). However, despite these attempts, the rate of missing scheduled appointments may still be high. While this rate was 12.8% in patients with health insurance and better economic status, it was 20.5% in patient groups with a poorer economic status (*Nayor, Maniar & Chan, 2017*; *Childers et al., 2016*). Additional research is needed to better understand and identify potential non-attenders.

Most previous research has focused on colonoscopy appointment adherence. A limited number of studies investigated non-adherence to all endoscopic procedures (*Wong, Zhang & Enns, 2009*; *Chang, Sewell & Day, 2015*). Therefore, an evaluation of non-attendance factors, taking into account procedure type, is warranted. Additionally, travel distance to the hospital, season of referral, and appointment lead times may be barriers to keeping appointments. Moreover, the widespread use of interventional endoscopic procedures and the demand for deep sedation during endoscopy have increased (*Predmore et al., 2017*). However, there are very few data on patients' compliance with procedure appointments performed under deep sedation. Finally, during the COVID-19 pandemic, both healthcare workers and patients have had additional concerns about viral infection while endoscopy units typically restricted endoscopic procedures to those who needed them most (*Hennessy et al., 2020*). The data on attendance to appointments during this period are absent, even though the efficient use of healthcare resources became more important than ever.

In this study, there were two aims. First, we aimed to examine the factors that affect non-attendance at endoscopy unit appointments. Second, we sought to explore the change, if any, in non-attendance rate during the COVID-19 outbreak.

## METHODS

### Study design and population

This study was a retrospective cohort study conducted at Kocaeli University Hospital, Kocaeli, Turkey. We examined non-attendance at gastroenterology endoscopy unit appointments in two different periods: January 2019–January 2020 (pre-pandemic) and March 2020–March 2021 (pandemic). The first 2 months of the 1-year pandemic period (March and April 2020) constituted the first lockdown in Turkey. During the following 7 months, there was no stay-at-home order, while in the final 3 months of the second study period, there was a second lockdown. Attendance at appointments was also separately analyzed according to lockdown periods.

Our endoscopy unit accepted referral patients from all subspecialties, including primary and secondary care providers. Administrative staff generated an electronic appointment record for patients who applied with an appointment request. Information about the appointment and instructions for the procedure was given verbally by doctors and also provided by administrative personnel in writing. As some endoscopy staff members were transferred to COVID-19 wards, the number of appointments was reduced by up to 30%, depending on the peaks of viral infection rates (*Bhandari et al., 2020*). While endoscopists were able to give moderate sedation, deep sedation was given by an anesthesia team, according to international guidelines (*Early et al., 2018*).

Patients over 18 years old who were scheduled for an appointment for intervention at the endoscopy unit were included in the study. Since we only participated in the endoscopic procedures of pediatric patients as consultants in emergencies, these patients were excluded from the study. Demographic data and postal addresses of duplicated patients with multiple admissions were most up-to-date in their latest appointment electronic data. Therefore, we have included only the last appointment of a patient with multiple appointments regardless of the results of the appointment. Liver biopsies were also excluded from endoscopy appointments due to the nature of the procedure. As emergencies and inpatient procedures were performed on the same day as the request, there was no waiting for an appointment. In addition, there were no travel restrictions for hospital staff and students staying on the university campus (travel distance < 4 km) and so these patient data were also excluded.

### Definition of variables

The main outcome was not attending the scheduled appointment. "Non-attenders" were defined as those who did not show up for the procedure and did not inform the department or patients who cancelled the appointment within 48 h from the scheduled appointment time. If the patient did not cancel the procedure 48 h before the procedure, these patients were evaluated as no-show, since a new appointment could not be created. "Attenders" were patients who completed the scheduled appointment. In our facility, despite communication channels, patients mostly just never showed up, and they infrequently cancelled their appointments.

Groupings of procedures were defined as follows. The endoscopic retrograde cholangiopancreatography (ERCP) procedure was classified as a single procedure group. Endoscopic ultrasonography and oral double-balloon enteroscopy were assessed in the gastroscopy group, while anal double-balloon enteroscopy was included in the colonoscopy group. Screening procedures are not routine or standard in our country. However, we perform surveillance and screening gastroscopies in patients with atrophic gastritis, intestinal metaplasia or a strong family history of gastric cancer. Repeat and sequential patient appointments were identified. The appointment outcomes of patients who had more than two appointments were also analyzed separately. Travel distance in kilometers (km) and time in minutes (min) between the endoscopy unit and the patient's postal address were calculated using Google Maps (Alphabet Inc., Mountview, CA, USA).

## Statistical analysis

In this study, patient-related factors (gender, age, marital status, associated malignancy, number of appointments, traveling status, travel time), hospital-related factors (premedication, referring physician), and healthcare-related factors (pre-pandemic or pandemic era, referral season, lead time, interventions) were considered independent variables while the non-attendance of the scheduled appointment was evaluated as a dependent variable. A quantile-quantile (q-q) plot was used to investigate the normality instead of the Kolmogorov–Smirnov test since the sample size was too large. Then, the independent samples t-test or Mann–Whitney U test was used to compare the attenders and non-attenders with respect to numeric variables (age, lead time and travel time). To compare the proportions of non-attendance for categorical variables (era, gender, marital status, interventions, referring physician, premedication, number of appointments, referral season, travelling status and associated malignancy), the Chi-square test was used. Lastly, simple and multiple logistic regression analyses were performed to determine the effect of independent variables on the dependent variable. Variables that were statistically significant in the univariate analysis, as well as other clinically important variables (age, gender, marital status), were assessed in multivariable regression analysis. All statistical analyses were performed using IBM SPSS for Windows, version 20.0 (IBM Inc., Armonk, NY, USA) and R 3.0.3 software (Institute for Statistics and Mathematics, Vienna, Austria). A $p$-value < 0.05 was considered statistically significant.

## Research ethics

The study protocol was reviewed and approved by the Kocaeli University Ethical Committee for Clinical Research (Identifier: GOKAEK-2021/4.26, Project No: 2021/79). Informed consent was not required because the study design was retrospective.

## RESULTS

### Study population

In the 12 months prior to and 12 months after the start of the COVID-19 pandemic, within a 24-month period, 11,690 appointments were initially identified. As a result of applying the exclusion criteria, the final study cohort was made up of 5,938 patients

(Fig. 1). Overall, in the final study group, 4,850 patients (81.7%) attended their appointments, and 1,088 (18.3%) did not. The mean (SD) age was 54.1 (SD 15.4) years, with roughly half of the study population, 3,025 (50.9%), being women and 4,638 (78.1%) being married (Table 1).

## Characteristics of patients scheduled for a procedure in the endoscopy unit

There were significant differences in univariate analyses between attenders and non-attenders. The overall proportion of non-attenders was higher before the COVID-19 pandemic compared to the pandemic era (22.6% $vs.$ 11.6%, $p < 0.001$) (Table 1). Moreover, during the pandemic, the non-attender rate was greater in the lockdown period as compared other times during the pandemic (16.9% $vs.$ 8.5%, $p < 0.001$).

There was no statistical difference ($p = 0.068$) in the mean (SD) age of attenders 53.9 (SD 15.3) years and non-attenders 54.9 (SD 15.8) years. However, non-attenders were more often referred by a physician other than a gastroenterologist (29.7% $vs.$ 14.4%; $p < 0.001$). Comparing the interventions, significantly more patients scheduled for ERCP (95.7%) attended compared to colonoscopy (81.7%) and gastroscopy (77.1%) ($p < 0.001$ for all).

The median interquartile range (IQR) lead time (in days) for attenders was found to be lower than that for non-attenders; 19 (9–33) $vs.$ 32 (21–42) ($p < 0.001$). A total of 1,668 patients were scheduled for more than one appointment, and 4,270 were scheduled for a single appointment. The non-attendance proportion was higher in patients with multi-scheduled appointments compared to those with single appointments (20.0% $vs.$ 17.7%; $p = 0.037$). Regarding appointment season, the proportion of non-attenders was significantly higher in winter than in summer (21.2% $vs.$ 16.4%, $p = 0.004$), fall (21.2% $vs.$ 16.7%, $p = 0.010$), and spring (21.2% $vs.$ 19.3%, $p = 0.241$).

No significant difference was observed between attenders and non-attenders in regard to the median (IQR) travel distance (kilometers) and the median (IQR) travel time (minutes) for all patients ((266 (155–373) $vs.$ 300 (187–359) $p = 0.250$) and (276 (165–383) $vs.$ 310 (196–369) $p = 0.239$), respectively). Interestingly, patients traveling from within the city did not attend appointments at a higher rate than patients traveling from outside the city (18.8% $vs.$ 16.3%, $p = 0.041$).

In terms of indications for procedures, non-attendance was highest for the pre- and post-pandemic gastroscopic screening procedures (33.7% and 32.6%, respectively) and lowest for pre-pandemic therapeutic ERCP procedures (3.0%) (Table 2). It should be noted that appointments for screening procedures were not scheduled during the pandemic period. However, some screening appointments we planned in the pre-pandemic period and coincided with the pandemic period. The highest rate of non-attendance was found for this group of appointments (Fig. 2).

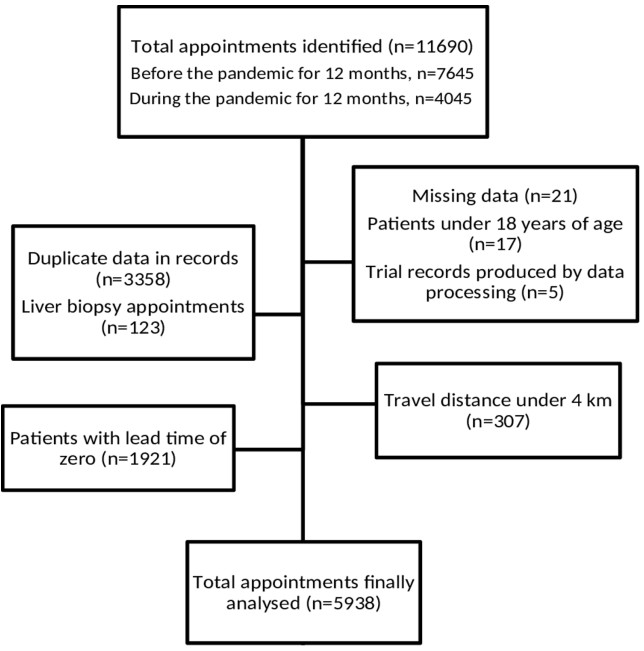

**Figure 1 Flowchart of the study population.**     

## Predictors of non-attendance at a scheduled procedure in the endoscopy unit

Univariate regression analysis identified female gender, more than one scheduled appointment, travelling to hospital from inside the city, absence of associated malignancy, absence of deep sedation, referral other than by a gastroenterologist, appointment during the COVID-19 pandemic era, longer lead time for gastroscopic and colonoscopic procedures, and referral season of Summer (when winter was the reference season) as factors significantly affecting attendance at appointments.

The independent predictors of non-attendance were identified using multiple logistic regression analyses. The estimated total effects of the variables were presented in Table 3 (*Westreich & Greenland, 2013*). Absence of deep sedation generated the highest odds ratio for non-attendance (odds ratio (OR): 3.253; 95% confidence interval (CI) [2.386–4.435], $p < 0.001$). Female gender, lead time, and referral by a physician other than a gastroenterologist were also significant independent predictors of non-adherence, while travel distance and season of appointment became non-significant. Lastly, patients scheduled for gastroscopy and colonoscopy procedures were less likely to attend appointments compared to those coming for an ERCP. Appointment during the COVID-19 pandemic, travel status, and marital status were not identified as significantly affecting non-attendance in regression analyses. The probability of non-attendance increased as the number of cumulative identified risk factors affecting a patient increased (Fig. 3).

## DISCUSSION

Failure to attend appointments is the first barrier to overcome in enhancing an endoscopy unit's productivity and efficiency. This study contributes to the literature by comparing

**Table 1 Characteristics of patients and appointment attendance.**

| Factors | Total (*n* = 5,938) | Non-attenders (*n* = 1,088) | Attenders (*n* = 4,850) | *p* |
|---|---|---|---|---|
| Patient-related factors | | | | |
| Gender | | | | |
| Male, *n* (%) | 2,913 (49.1) | 482 (16.5) | 2,431 (83.5) | 0.001[a] |
| Female, *n* (%) | 3,025 (50.9) | 606 (20) | 2,419 (80) | |
| Age, mean (SD), years | 54.1 (15.4) | 54.9 (15.8) | 53.9 (15.3) | 0.068[b] |
| Marital status | | | | |
| Married, *n* (%) | 4,638 (78.1) | 834 (18) | 3,804 (82) | 0.209[a] |
| Not married, *n* (%) | 1,300 (21.9) | 254 (19.5) | 1,046 (80.5) | |
| Number of appointments | | | | |
| Single, *n* (%) | 4,270 (71.9) | 754 (17.7) | 3,516 (82.3) | 0.037[a] |
| Multiple, *n* (%) | 1,668 (28.1) | 334 (20) | 1,334 (80) | |
| Travelling status | | | | |
| From the city, *n* (%) | 4,740 (79.8) | 893 (18.8) | 3,847 (79.2) | 0.041[a] |
| From outside the city, *n* (%) | 1,198 (20.2) | 195 (16.3) | 1,003 (83.7) | |
| Travel time median (IQR), min | 277 (167–381) | 310 (196–369) | 276 (165–383) | 0.239[c] |
| Associated malignancy | | | | |
| Yes, *n* (%) | 2,012 (33.9) | 220 (11) | 1,792 (89) | <0.001[a] |
| No, *n* (%) | 3,926 (66.1) | 868 (22) | 3,058 (78) | |
| Hospital-related factors | | | | |
| Deep sedation | | | | |
| Yes, *n* (%) | 1,351 (22.8) | 63 (4.7) | 1,288 (95.3) | <0.001[a] |
| No, *n* (%) | 4,587 (77.2) | 1,025 (22.3) | 3,562 (77.7) | |
| Referring physician | | | | |
| Gastroenterologist, *n* (%) | 4,403 (74.1) | 632 (14.4) | 3,771 (85.6) | <0.001[a] |
| Other, *n* (%) | 1,535 (25.9) | 456 (29.7) | 1,079 (70.3) | |
| Healthcare-related factors | | | | |
| Era | | | | |
| Before pandemic, *n* (%) | 3,618 (60.9) | 818 (22.6) | 2,800 (77.4) | <0.001[a] |
| After pandemic, *n* (%) | 2,320 (39.1) | 269 (11.6) | 2,051 (88.4) | |
| Lead time, median (IQR), days | 21 (11–35) | 32 (21–42) | 19 (9–33) | <0.001[c] |
| Interventions | | | | |
| ERCP, *n* (%) | 868 (14.6) | 37 (4.3) | 831 (95.7) | <0.001[a] |
| Gastroscopy, *n* (%) | 2,674 (50.5) | 613 (22.9) | 2,061 (77.1) | |
| Colonoscopy, *n* (%) | 2,396 (40.4) | 438 (18.3) | 1,958 (81.7) | |
| Referral Season | | | | |
| Spring, *n* (%) | 1,320 (22.2) | 255 (19.3) | 1,065 (80.7) | 0.001[a] |
| Summer, *n* (%) | 1,563 (26.3) | 256 (16.4) | 1,307 (83.6) | |
| Fall, *n* (%) | 1,552 (26.1) | 259 (16.7) | 1,293 (83.3) | |
| Winter, *n* (%) | 1,503 (25.3) | 318 (21.2) | 1,185 (78.8) | |

**Notes:**
[a] Chi-squared test.
[b] Student's t-test.
[c] Mann-Whitney U test. *P* > 0.05 in the listed factors indicates that there is no difference between the attender and non-attender groups.

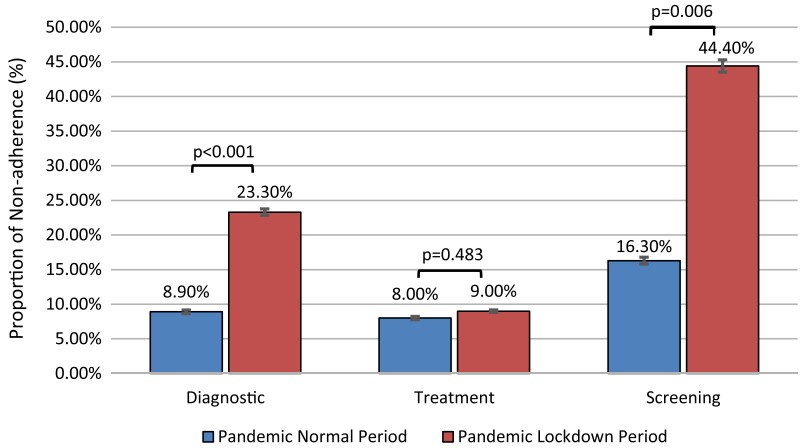

**Figure 2** Proportion of non-attendance for gastrointestinal intervention during COVID-19 with and without lockdown by indication.

**Table 2 Comparison of adherence to GI procedures during pre-pandemic and pandemic eras.**

| | Pre-pandemic | | | Pandemic | | |
|---|---|---|---|---|---|---|
| | Non-attenders, *n* (%) | Attenders, *n* (%) | *p* | Non-attenders, *n* (%) | Attenders, *n* (%) | *p* |
| Gastroscopy | | | <0.001 | | | 0.084 |
| Diagnostic | 117 (16.7) | 584 (83.3) | | 88 (19.1) | 372 (80.9) | |
| Therapeutic | 27 (10.1) | 240 (89.9) | | 71 (21.8) | 254 (78.2) | |
| Screening | 295 (33.7) | 580 (66.3) | | 15 (32.6) | 31 (67.4) | |
| Colonoscopy | | | <0.001 | | | <0.001 |
| Diagnostic | 108 (18.7) | 470 (81.3) | | 18 (9.1) | 179 (90.9) | |
| Therapeutic | 43 (13.0) | 289 (87.0) | | 32 (6.5) | 463 (93.5) | |
| Screening | 221 (29.7) | 522 (70.3) | | 16 (31.4) | 35 (68.6) | |
| ERCP | | | 0.005 | | | 0.010 |
| Diagnostic | 5 (22.7) | 17 (77.3) | | 6 (12.0) | 44 (88.0) | |
| Therapeutic | 3 (3.0) | 97 (97.0) | | 23 (3.3) | 673 (96.7) | |

**Note:**
GI, Gastrointestinal; ERCP, Endoscopic Retrograde Cholangiopancreatography.

pre-pandemic and pandemic attendance frequencies and defining the predictors of non-attendance at endoscopy unit appointments.

We identified the absence of deep sedation as a strong predictor of non-attendance at an endoscopic procedure appointment. Furthermore, gastroscopy and colonoscopy, referral by a physician other than a gastroenterologist, female gender, and lead time were other independent predictors of non-attendance. However, marital status, the effect of the COVID-19 pandemic, and travel status were not independent predictors of compliance.

The scarcity of healthcare resources during the COVID-19 outbreak has increased the importance of compliance with appointments. We found an overall non-attendance rate (18.3%) for the endoscopy unit that was compatible with the literature (*Chang, Sewell & Day, 2015*; *Shuja et al., 2019*; *Nayor, Maniar & Chan, 2017*). During the COVID-19 period, patient non-attendance rates actually improved significantly, falling to 11.6% compared

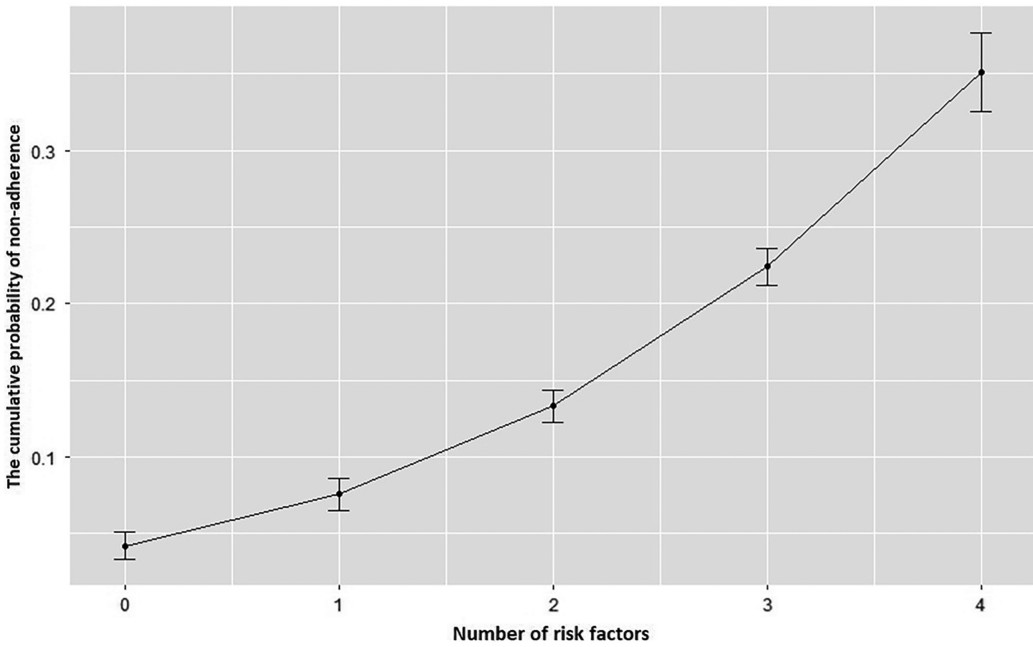

**Figure 3** **The cumulative probability curve of non-attendance at appointments by the number of risk factors present in a patient.** The probability of non-adherence increased cumulatively as the number of risk factors present in any individual patient increased. The cumulative risk factors assessed were: female gender, intervention other than ERCP, referral other than a gastroenterologist, absence of deep sedation.

with 22.6% pre-COVID-19 ($p < 0.001$). During this period, especially at the beginning of the pandemic, many units severely limited the number of appointments. They created endoscopy unit appointment priorities (*Hennessy et al., 2020*). Therefore, patients may have been more reluctant to miss an appointment because it was difficult to obtain a new one.

Unsurprisingly, the highest non-attendance rates for GI endoscopy appointments were recorded for screening procedures during the COVID-19 era. In contrast, the highest level of participation was evident for therapeutic procedures during the lock-down period. Lock-down periods have their own special limitations, such as 'stay at home orders', which decrease patients' ability to access healthcare. Apart from these limitations, the indication for the procedure influenced the attendance at appointments. Thus, the highest level of participation was seen for therapeutic and diagnostic procedures and the lowest for screening procedures in all periods. Previous studies have also shown that attendance at GI endoscopy appointments is lower for screening procedures (*Shuja et al., 2019*; *Greenspan et al., 2015*) and higher for diagnostic and therapeutic procedures (*Wong, Zhang & Enns, 2009*). These findings are relevant for preventive medicine. Endoscopy units should develop strategies to increase compliance with screening procedures, especially in the COVID-19 era. Otherwise, the incidence of GI diseases presenting at more advanced stages may increase during the COVID-19 pandemic, which continues.

An interesting and novel finding of this study was that patients who underwent endoscopic procedure with planned deep sedation had a higher rate of compliance with

**Table 3 Individual and organizational factors associated with non-attendance: ORs and 95% CIs (n = 5,938).**

| Factors | Univariate analysis | | Multivariate analysis | |
|---|---|---|---|---|
| | OR (95% CI) | p | OR (95% CI) | p |
| Patient-related factors | | | | |
| Gender | | | | |
| Male (R) | 1.0 | | 1.0 | |
| Female | 1.263 [1.107–1.442] | 0.001 | 1.187 [1.033–1.363] | 0.015 |
| Age, years | 1.004 [1.000–1.008] | 0.063 | – | – |
| Marital Status | | | | |
| Married (R) | 1.0 | | 1.0 | |
| Not married | 1.108 [0.947–1.295] | 0.200 | 1.169 [0.993–1.377] | 0.061 |
| Number of appointments | | | | |
| One time (R) | 1.0 | | – | – |
| More than one | 0.488 [0.410–0.582] | <0.001 | | |
| Travelling Status | | | | |
| In the city (R) | 1.0 | | 1.0 | |
| Outside the city | 0.838 [0.707–0.993] | 0.041 | 0.935 [0.782–1.117] | 0.459 |
| Travel time | 1.000 [1.000–1.001] | 0.613 | – | – |
| Associated Malignancy | | | | |
| Yes (R) | 1.0 | | – | – |
| No | 2.312 [1.972–2.711] | <0.001 | | |
| Hospital-related factors | | | | |
| Deep sedation | | | | |
| Yes (R) | 1.0 | | 1.0 | |
| No | 5.883 [4.526–7.647] | <0.001 | 3.253 [2.386–4.435] | <0.001 |
| Referring physician | | | | |
| Gastroenterologist (R) | 1.0 | | 1.0 | |
| Other | 2.522 [2.196–2.895] | <0.001 | 1.891 [1.630–2.193] | <0.001 |
| Healthcare-related factors | | | | |
| Era | | | | |
| Before pandemic (R) | 1.0 | | 1.0 | |
| After pandemic | 0.448 [0.386–0.520] | <0.001 | 1.010 [0.847–1.203] | 0.913 |
| Lead time, days | 1.007 [1.009–1.011] | <0.001 | 1.006 [1.004–1.008] | <0.001 |
| Interventions | | | | |
| ERCP (R) | 1.0 | | 1.0 | |
| Gastroscopy | 6.680 [4.748–9.399] | <0.001 | 2.213 [1.495–3.277] | <0.001 |
| Colonoscopy | 5.024 [3.557–7.096] | <0.001 | 1.819 [1.227–2.697] | 0.003 |
| Referral Season | | | | |
| Spring (R) | 1.0 | | – | – |
| Summer | 0.818 [0.676–0.991] | 0.040 | | |
| Fall | 0.837 [0.691–1.013] | 0.067 | | |
| Winter | 1.121 [0.932–1.348] | 0.225 | | |

**Note:**
OR, odds ratio; CI, confidence interval; R, reference.
their appointments. However, moderate sedation could be safer, and the recovery period of these patients would be relatively short. Gastroenterologists can give moderate sedation without the support of anesthetists in many countries (*Early et al., 2018*). Despite these advantages, the comfort of planned deep sedation appears to be important to patients and increased the rate of attendance in our population. Anxiety about endoscopic procedures may be significant obstacles for patients attempting to adhere to their appointments. *Shafer et al. (2018)* reported that 29% of high anxiety scores regarding colonoscopy were associated with the procedure itself. In another study, 46–52% of patients who missed or canceled their appointments were anxious that they would undergo an unpleasant experience during the endoscopic procedure (*Bhise et al., 2016*). Planned deep sedation may counter the high rates of patient anxiety associated with an endoscopic procedure and, consequently, increase attendance rates.

We found that patients referred by physicians other than gastroenterologists were less likely to adhere to appointments. Three previous studies investigating the effect of referral source on adherence to appointments reported conflicting findings. *Sola-vera et al. (2008)* and *Adams, Pawlik & Forbes (2004)* found that patients referred by a gastroenterologist had a higher rate of attending an appointment as compared to those referred by general practitioners (GPs) or other physicians. In contrast, *Wong, Zhang & Enns (2009)* compared non-attendance among patients referred by GPs and other specialists to the endoscopy unit and found that those referred by GPs were more likely to keep their appointments. Our results were in keeping with the findings of *Sola-vera et al. (2008)* and *Adams, Pawlik & Forbes (2004)* This may be because, as a group of physicians who will personally perform the procedures, gastroenterologists may be better at conveying the importance and necessity of the procedure to the patients and thus motivate them to adhere to their appointments. Furthermore, gastroenterologists are the best source of information concerning these types of procedures and so they may best meet the patient's informational needs regarding the procedure. Likewise, *Lloyd, Bradford & Webb (1993)* previously found that not attending gastroenterology endoscopy unit appointments was associated with an inability to discuss health problems with GPs. Providing sufficient information about the procedure and motivating patients might improve adherence to appointments.

Compliance with appointments for advanced procedures, such as ERCP, was higher than for colonoscopy and gastroscopy. Physicians may perceive interventional procedures as a priority as compared to diagnostic procedures and thus more successfully motivate patients to attend. In addition, advanced procedures are performed less frequently at many centers and are difficult to schedule. Once patients are scheduled to attend, they are possibly more motivated to attend an appointment that is more difficult to reschedule compared to more widely available endoscopic procedures. *Chang, Sewell & Day (2015)* also found that undergoing more advanced endoscopic procedures was inversely and independently associated with non-attendance.

We found longer lead time (*Blumenthal et al., 2015*) and female gender (*Denberg et al., 2005*) were both independent risk factors for poor adherence, as shown in previous studies. In parallel with previous research, we also confirmed that a higher proportion of patients did not attend appointments during Winter (*Smith et al., 2020*), with multi-scheduled

appointments (*Torres et al., 2015*), and patients with malign comorbidity (*White et al., 2021*) by univariate analysis. However, these variables lost significance in multivariate analysis in our population. These parameters have previously been associated with non-attendance, though only inconsistently (*Shuja et al., 2019*; *Blæhr et al., 2016*; *Wolff et al., 2019*; *Guay et al., 2014*). In light of these findings, logistic and social support may decrease the non-attendance rates of those with comorbidities and those traveling to the hospital during Winter. A longer lead time can cause non-adherence to appointments, leading to multi-scheduling, which in turn leads to a higher likelihood of non-attendance and thus longer lead times. Endoscopy unit administrative personnel should target these modifiable factors in order to enhance adherence to appointments.

Our study has certain limitations. First, there was a gap in our data collection period. Due to the incompatibility between the emergence of COVID-19 cases in our country and the transition to pandemic conditions, we did not include the remaining month (February) in either period. Second, our study was retrospective in nature, and selection bias is possible. During the pandemic, screening procedures were not prioritized for an appointment while making appointments for mostly urgent cases that could not be postponed, and this may have led to further selection bias. Our study could include possible confounding variables. The estimates of variables provided for non-attendance to endoscopy appointments in the multivariate logistic regression were total effects rather than direct effects. Therefore, confounding bias cannot be ignored in our study (*Westreich & Greenland, 2013*). Despite these limitations, our electronic database was reliable and multiple appointments were eliminated. Our center is a tertiary referral center, we accept referrals from primary and secondary tertiary healthcare facilities and all subspecialties reflecting a broad population. Also, we accomplished strict exclusion criteria allowing accurate analysis. Thus, we have eliminated some threats to external validity. Identification of COVID-19 specific factors, such as fear of infection, curfew, and economic impact that may affect compliance with gastroenterology endoscopy unit appointments were not the aim of the present study and should be addressed in future studies.

In conclusion, access to gastrointestinal endoscopic procedures is limited in many parts of the world, especially during the COVID-19 pandemic. It is vital that patients adhere to these appointments to ensure the proper utilization of the limited healthcare resources. Our results showed that female patients, those undergoing endoscopic procedures without deep sedation, those referred by physicians other than gastroenterologists, those with longer waiting list times, and patients scheduled for screening procedures, especially during the COVID-19 period, were less likely to adhere to the appointments. Endoscopy units should develop strategies to increase attendance to appointments, especially for those who have cumulative risk factors.

## ACKNOWLEDGEMENTS

We are grateful to Dr. Sibel Balci of the Biostatistics and Medical Informatics Department of Kocaeli University for his support in the statistical analysis. We would also like to thank Mr. Jeremy Jones of the Academic Writing Department of Kocaeli University for his assistance in editing the English used in this article.

### Funding
The authors received no funding for this work.

### Competing Interests
The authors declare that they have no competing interests.

### Author Contributions

- Hasan Yılmaz conceived and designed the experiments, performed the experiments, analyzed the data, prepared figures and/or tables, authored or reviewed drafts of the article, and approved the final draft.
- Burcu Kocyigit performed the experiments, analyzed the data, prepared figures and/or tables, authored or reviewed drafts of the article, and approved the final draft.

### Human Ethics

The following information was supplied relating to ethical approvals (*i.e.*, approving body and any reference numbers):

The Kocaeli University Ethical Committee of Clinical Research reviewed and approved the study (Project No: 2021/79).

### Data Availability

The raw data of endoscopy unit adherence is available in the Supplemental Files.

### Supplemental Information

Supplemental information for this article can be found online at http://dx.doi.org/10.7717/peerj.13518#supplemental-information.

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
