# Peer review of "Factors associated with non-attendance at appointments in the gastroenterology endoscopy unit: a retrospective cohort study"

_PeerJ, doi:10.7717/peerj.13518_

## Round 0.1 · original submission · Major Revisions

From the Editor:

Overall:
Please improve the quality of English comprehension.

Methods:
State the name of tertiary center with city and country of location.
Mention the parent company name and location in brackets with regard to Google Maps.

Results:
A good number of patients were excluded from the initial sample data.
What about seasonal attendance in Spring? Is that a season seen in the city where the study is performed?
172. What does screening gastroscopy refer to? Not sure what the indication for screening is? Is this a standard of practice in the country where the study was performed?
173-176. Revise this sentence, unclear what the authors are stating.
177. Rephrase the heading
179-187. This entire paragraph is poorly written in terms of quality of English used. Please ask a native English speaker to review and re-write it.

Discussion: Several errors in writing noted. Please make it more concise and have a native English speaker review the discussion.

In Figure 1, the authors state 3.358 had duplicate records, please correct that number and remove the “Dot”.

·

Basic reporting

The authors addressed very important questions about which are the factors associated with non attendance to endoscopic procedures. The article is very well written and I think this kind of articles allow us to reflect on parts of the health system that need to be improved.
This study has the objective to identify factors associated with non attendance to an endoscopic procedure. I think that the title should reflect the objective of the study and not the results of which factors were associated with non attendance in the study. I think a title like "factors associated with non attendance to... in ..." would be more appropriate as a title for this study.
The factors associated with non attendance are different considering different appointments, contexts and populations. Please specify in the introduction if all factors associated with non attendance are associated with non attendance to endoscopic procedures or not.
What are the differences in the non attendance rates from 3.6 to 67% in line 52 and from 12.8 to 20.5 in line 64? PLease add additional information to understand the differences between these 2.

Experimental design

In line 112 it says "Recurrent patient appointments were identified, and such patients’ last appointments were analyzed" why last appointment? why not first or all of them? I think it would be important to add a clarification here.
in line 115, Cancellations near the appointment within 48 hours were considered as non attendance. Non attendance and cancellation are usually very different conducts and usually do not have the same impact in the health systema. Could you add additional information for this decision?
In line 141 it says that after excluding inconsistent data, they analyzes 5938/11690. Almost have of the appointments were excluded becouse inconsistent data? what does this mean? >What was the definition of inconsistent data? how does this high exclusion rate affects the validity of the results?
Table 1. I suppose the p values correspond to the null hypothesis of equal distribution between atenders and non attenders. I would suggest changing the order of the columns and leaving them to total first. I would also suggest clarifying as a footnote what general hypothesis the p values are testing.
In table 1 all percentages are given for columns and not for rows. That does not allow to tell which is the percentage of nonattenders among men, or during the pandemics. I think this should be changed to the porcentaje within rows.
In table 1, please review ".001" if it corresponds to 0.001 or to <0.001
In table 1, it is extrange the way numeric variables are presented with mean and standard deviation in the "n" column and the min and max under the "%" column. Maybe this last information can be avoided.
In table 1, please provide the unit of quantitative variables like age and lead time.
The percentage in line 150 for example are discordant with the percentages presented in table 1. That is becouse of the difference between within row percentages and within column percentages. I would suggest to be homogeneous with this numbers. Probably it would be more informative to show percentages of non attendance with the different categories.
Probably it would be more clear not to use abbreviations for attenders and non attenders.
In line 154, I believe it is more cler to use standard deviation instead of +- symbol. Probably 1 decimal is more clear for these numbers.
In line 156 the porcentages 29 and 14 are not presented in table 1. This is becouse of the same problem if the authors should present within row or within column porcentajes. I believe that the more important porcentajes might be the percentage of non attenders within each category.
Please add the units for the travel distances in line 167
Please remove the extra "." in "21.2." in line 164
In my opinion, the part of the predictors of non attendance is the most important part of the results. I think it would be very important to provide all the OR of each relationship for each potential characteristic and non attendance. This is in my opinion much more important than the comparison in proportion provided in the previous section, that of course is also important. The association measurement provides much more information than p values.
In this particular part, I believe that there is some confusion. If the main objective of the study is to identify predictors, I think the authors should present bivariate association and avoid presenting multivariate association measurements (like using multivariate models).
If the authors objective is to generate predictive models to non attendance, I would suggest to follow the standard procedures to generate and validate predictive models like the method described by Towards better clinical prediction models: seven steps for development and an ABCD for validation Ewout W. Steyerberg* and Yvonne Vergouwe
I think it is very important to define if the objective of the paper is to identify potential predictors, or to generate and validate a predictive model. Because that would help to complete the results, methods and discussion section.
In table 2, there seem to be many factors missing. I think it would be important to complete the table with all missing factors. I believe that OR estimators should be bivariated and not multivariate to avoid table 2 fallacy.
Figure 1. Why is a distance under 4 km excluded? Is there a reason for that?
Figure 2. I think it is better to change the figure to a 2d figure. Usually 3d figures are less clear and more difficult to interpret
Figure 3 seems to assume that the number of risk factors to non attendance is related to the probability of non attendance. Since the effect of each factor may be very different, this figure probably needs additional clarifications.

Validity of the findings

In the discussion, I think it is very important to add that probably there is a selection bias affecting those endoscopies appointed during the pandemics since they may be reduced to only those procedures that, considering the nature of the condition, could not be postponed until the pandemics have gone. For example, maybe most screening studies could be delayed without consequences.
I think the interpretation in line 220 about sedation is at least mischieving. It is possible that the different types of sedation are related to different procedures. This interpretation should be taken with caution, since the nature of the study is not defined to address causality. The danger part of this kind of interpretation is to promote one kind of sedation to decrease non attendance rate. This kind of interpretations should be taken with caution.

Additional comments

-

Reviewer 2 ·

Basic reporting

The writing appears to be clear and relative straightforward. There are a few minor issues with grammar and sentence construction which can be easily corrected. These are a few examples:

Line 72-74: The last word of that sentence should be infection rather than "contamination". Humans get infected with viruses while inanimate objects get contaminated.

Line 97-99: ....deep sedation "was applied with by".... The use of 2 prepositions back to back here is wrong. You either "apply with" or "apply by". In this context, "administered by" or "given by" seems more appropriate.

Line 89-91: I'm not sure what distinction the authors are trying to make here. The authors claim they intended to "explore the parameters associated with non-attendance to endoscopy appointments" and then go ahead to say "factors associated with Covid-19 was subject of another study". It appears this study also focused on appointments in Covid-19. In fact, about half of the study period was during Covid-19. A better clarification of what the authors mean by this statement is needed to avoid a potential Salami publication where similar data is presented as 2 or more separate publications.

Experimental design

The study design appears adequate, although the research question appears to have been previously explored in previous publications as cited by the authors with the major distinction in this article being the effect of the pandemic and deep sedation on appointments compliance. Despite being a single center study, this research provides a window to understanding how the pandemic affects utility of healthcare especially as it relates to GI procedures and helps identify target opportunities to improve or address non-attendance

Validity of the findings

With the aid of logistic regression, the authors identify various predictors of nonattendance including absence of deep sedation, female gender, type of procedure and referral physician. In the discussion, the authors go on to say "winter season, multi-scheduled appointments, and malign comorbidity" were associated with non-attendance (Line 257-258). These were however not mentioned in the earlier report of Predictors (Line 177-187). These variables were also not presented in Table 3 or referenced to any other table or figure. How did the authors arrive at these other predictors? They should be mentioned in the section 3.3 (Predictors of Nonattendance) and either included in Table 3.

Additional comments

Line 172 mentions "screening procedures of gastroscopy". Can the authors enlighten the readers what conditions are being screened for as screening gastroscopies is not universal like screening colonoscopies.

Overall, the article helps identify potential opportunities for intervention to reduce nonattendance. The finding of the impact of deep sedation is particluarly revealing. We need to do a better job at reassuring patients, allaying their fears and educating them on the use of conscious sedation. We also need to refine out technique to potentially reduce patient discomfort.

·

Basic reporting

I have suggested considering the formatting of the title, the authors have provided major results in the title. I have a suggestion to please rewrite the title as Evaluation of Non-adherence to endoscopy appointments at a gastroenterology unit; A retrospective analysis

This manuscript is well structured, however; it lacks scientific writing at some points. I will suggest considering careful proof read after incorporating all the comments from reviewers.

The introduction section states that there is no study on this topic, but only colonoscopic data is available, can authors confirm their best search, and no other study, nationally or internationally has been done on this objective.

Experimental design

Can the authors explain why appointments for patients with age less than 18 years were not included in the study?

The heading no. 2 should only be "Methods".

Can authors explain on which grounds the variables were selected for a regression analysis? were they reported previously in other studies, or were they logically plausible with non-adherence, or did authors consider all variables to be run in univariate analysis? Please write dependent and independent variables in the analysis section

There is to define some terms in methodology i.e. what is meant by non-adherence? attenders etc.. were the attendees those who completed the procedure?

Validity of the findings

Authors have primarily compared between attenders and non-attenders and then evaluated various predictors of non-adherence to the procedure. Based on the rationale of the study, these non-adherence practices result in considerable financial loss as well as reduced trust/quality of patients. Authors have dug out various predictors, can they classify these predictors i.e. patient-related factors, hospital-related factors, healthcare-related factors. As gastroscopy is not related to the patient`s factor associated with non-adherence. Similarly, deep sedation is something related to the hospital.

There are many other factors that may associate with non-adherence i.e. comorbid conditions, psychological impairments, financial status, disease severity, health insurance status...... and so on. can author draw a firm conclusion without incorporating these factors in the model. Moreover, the exact cause of non-adherence is something more important than predictors. Authors should consider them while discussing the results.

---

## Round 0.2 · Minor Revisions

Dear Authors,

I think the quality of the manuscript has significantly improved but one of the reviewers raises an important point especially regarding exclusion criteria. Please see the comments below and respond accordingly.

Thanks


Basic reporting

I think the article improved significantly and all my concerns were clarified and improved significantly. In my opinion, the paper quality has substantially improved.

I agree with the exclusion criteria detailed in the responses point by point to the comment "In line 141 it says that after excluding inconsistent data, they analyze 5938/11690. Almost have of the appointments were excluded because inconsistent data? what does this mean? >What was the definition of inconsistent data? how does this high exclusion rate affects the validity of the results?". Please be sure these exclusion criteria are describe explicitly in the paper, allowing the reader to understand potential selection bias and validity of the results. The comment was directed to the external validity and not to the power of the study.


Validity of the findings

I still believe that the adjusted OR may represent here a Table 2 fallacy. Please see The Table 2 Fallacy: Presenting and Interpreting Confounder and Modifier Coefficients - Daniel Westreich and Sander Greenland. I think this can be acknowledged in the discussion, to interpret with caution the results of this adjusted OR if the authors decide to keep them in the table.

·

Basic reporting

I think the article improved significantly and all my concerns were clarified and improved significantly. In my opinion, the paper quality has substantially improved.
I agree with the exclusion criteria detailed in the responses point by point to the comment "In line 141 it says that after excluding inconsistent data, they analyze 5938/11690. Almost have of the appointments were excluded because inconsistent data? what does this mean? >What was the definition of inconsistent data? how does this high exclusion rate affects the validity of the results?". Please be sure these exclusion criteria are describe explicitly in the paper, allowing the reader to understand potential selection bias and validity of the results. The comment was directed to the external validity and not to the power of the study.

Experimental design

-

Validity of the findings

I still believe that the adjusted OR may represent here a Table 2 fallacy. Please see The Table 2 Fallacy: Presenting and Interpreting Confounder and Modifier Coefficients - Daniel Westreich and Sander Greenland. I think this can be acknowledged in the discussion, to interpret with caution the results of this adjusted OR if the authors decide to keep them in the table.

Additional comments

In my opinion, the paper quality has substantially improved.

·

Basic reporting

The authors have addressed all my concerns very well. I have no more comments on the manuscript.

Experimental design

The authors have addressed all my concerns very well. I have no more comments on the manuscript.

Validity of the findings

The authors have addressed all my concerns very well. I have no more comments on the manuscript.

Additional comments

The authors have addressed all my concerns very well. I have no more comments on the manuscript.

---

## Round 0.3 · accepted · Accept

Congratulation to the authors. Looks like the concerns of reviewers were adequately addressed.